# Knowledge Graph Compression
# Enhances Diverse Commonsense Generation

**EunJeong Hwang**[1,2], **Veronika Thost**[3], **Vered Shwartz**[1,2], **Tengfei Ma**[4]

[1] University of British Columbia   [2] Vector Institute for AI
[3] MIT-IBM Watson AI Lab, IBM Research
[4] Stony Brook University

{ejhwang,vshwartz}@cs.ubc.ca
veronika.thost@ibm.com
tengfei.ma@stonybrook.edu

## Abstract

Generating commonsense explanations requires reasoning about commonsense knowledge beyond what is explicitly mentioned in the context. Existing models use commonsense knowledge graphs such as ConceptNet to extract a subgraph of relevant knowledge pertaining to concepts in the input. However, due to the large coverage and, consequently, vast scale of ConceptNet, the extracted subgraphs may contain loosely related, redundant and irrelevant information, which can introduce noise into the model. We propose to address this by applying a differentiable graph compression algorithm that focuses on more salient and relevant knowledge for the task. The compressed subgraphs yield considerably more diverse outputs when incorporated into models for the tasks of generating commonsense and abductive explanations. Moreover, our model achieves better quality-diversity tradeoff than a large language model with 100 times the number of parameters. Our generic approach can be applied to additional NLP tasks that can benefit from incorporating external knowledge.[1]

## 1 Introduction

Commonsense knowledge graphs (CSKGs) have been used to improve the performance of downstream applications such as question answering (Yasunaga et al., 2021) and dialogue (Tu et al., 2022), as well as for enhancing neural models for commonsense reasoning tasks (Lin et al., 2019; Yu et al., 2022). Typically, these methods extract keywords from the input and construct a subgraph around them using the KG knowledge, which is then incorporated into the model.

Recent popular CSKGs such as ConceptNet (Speer et al., 2017) and ATOMIC (Sap et al., 2019) represent nodes in natural language, which allows flexibility but also adds redundancy and noise (Wu

---

[1]Code is available at:
https://github.com/eujhwang/KG-Compression

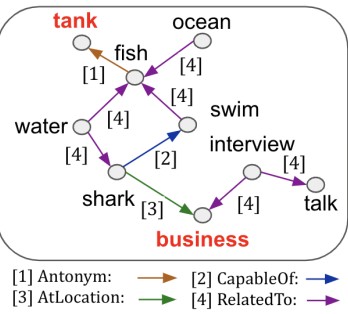

**Input:** A shark interviews a fish.

[1] Antonym: →   [2] CapableOf: →
[3] AtLocation: →   [4] RelatedTo: →

**References:**
1. Sharks and fish do not talk.
2. A shark cannot talk.
3. Fish cannot talk.

Figure 1: An example from ComVE (Wang et al., 2020). The subgraph obtained for the input sentence includes unimportant information (in red) that can lead to noisy outputs.

et al., 2023). Moreover, the retrieved subgraphs around a task's concepts potentially include information that is not relevant to the context. For example, in Figure 1, the goal is to generate a reason why the input sentence ("A shark interviews a fish") defies commonsense. The concepts `tank` and `business` are semantically irrelevant to either the input or the reference output sentences. Including irrelevant information introduces noise that can deteriorate the model's performance. Recent work has addressed this by pruning noisy paths based on low edge confidence scores in knowledge base embeddings (Lin et al., 2019) or by using language models (LMs) (Yasunaga et al., 2021). Yet, the relevance of paths is not determined *in relation to the given task*.

In this paper, we propose to use differentiable graph compression that enables the model to learn how to select the crucial concepts that are actually related to the task. Our method contains two main components: using self-attention scores to select relevant concept nodes in the retrieved subgraph,

and employing optimal transport loss to ensure the chosen concepts preserve the most crucial information of the original graph. In this way, the irrelevant or redundant concepts can be automatically eliminated in the subgraph.

We demonstrate the usefulness of our method on two commonsense generation tasks: commonsense explanation generation and abductive commonsense reasoning. Our method outperforms a range of baselines that use KGs in terms of both diversity and quality of the generations. We further conduct a comprehensive analysis, exploring a different setup, such as the scenario of incorporating new knowledge into the subgraph. Different from the baselines, our method enables the model to maintain performance, even in the presence of potentially increased noisy data. Finally, we show that our approach demonstrates better quality-diversity tradeoff than the large language model vicuna-13b, which has 100 times more parameters.

## 2 Background

**KG-Enhanced Neural Methods.** KGs have been used to enhance models for question answering (Lin et al., 2019; Feng et al., 2020; Yasunaga et al., 2021), relation classification (Wang et al., 2021), textual entailment (Kapanipathi et al., 2020), and more. Typically, such methods extract a subgraph of knowledge related to keywords in the input, which is then either embedded or represented in natural language before being incorporated into the model. For example, both Wang et al. (2023) and Wang, Fang, et al. (2023) used CSKGs to enhance a commonsense inference and a QA model by including the abstraction of concepts in the input (e.g. vacation → relaxing event). However, some knowledge may be irrelevant in the context of the particular question.

To reduce such noise, prior methods have proposed to score and prune the paths. Lin et al. (2019) used TransE (Wang et al., 2014) to score each edge in the path, while Yasunaga et al. (2021) scores nodes based on the likelihood of a pre-trained LM to generate it after the input. In both methods, the scores are not trained to represent a node's importance in relation to the task.

**Generating Commonsense Explanations.** This paper focuses on the task of generating commonsense explanations, in particular focusing on the following datasets. In ComVE (Wang et al., 2020) the goal is to generate explanations for why a given

sentence, such as "A shark interviews a fish", does not make sense. $\alpha$-NLG (Bhagavatula et al., 2020) presents models with a past observation, such as "Mike spends a lot of his time on the internet" and a future observation such as "Now other people love the internet because of Mike's website". The goal is to generate a plausible explanation for what might have happened in-between, such as "Mike created a website that helps people search". In a related line of work, researchers collected or generated commonsense explanations for existing tasks (e.g., Camburu et al., 2018; Rajani et al., 2019; Brahman et al., 2021).

**Diverse Sentence Generation.** One of the desired aspects of generating commonsense explanations is the diversity of the outputs. Popular LM decoding methods such as top-k (Fan et al., 2018), top-p (Holtzman et al., 2020), and truncated sampling (Hewitt et al., 2022) generate diverse outputs by pruning the probability distribution over the vocabulary for the next token and then sampling a token from the pruned distribution. An alternative approach is to use a mixture of experts (MoE) to produce diverse outputs (Shen et al., 2019; Cho et al., 2019). Our approach extends MoKGE Yu et al. (2022), a model for commonsense explanation generation. MoKGE uses a combination of KGs to diversify the outputs of a MoE model. However, the knowledge that MoKGE retrieves from the KG is not filtered, hence may contain loosely related, redundant and irrelevant information, which can negatively impact the model's performance in generating high-quality diverse outputs. In our approach, we employ knowledge graph compression to prioritize more important information.

## 3 Method

Our goal is to generate diverse sentences, $\{y_1, y_2, ..., y_k\}$ that explain a given instance $x$ (see Sec 2 for the specific task descriptions). The objective is to maximize the probability of generating each $y_i$: $P(y_i|x)$, as well as to diversify them. Previous KG-enhanced approaches usually add an external graph $\mathcal{G}_x$ to make the generation also conditioned on the graph: $P(y_i|x, \mathcal{G}_x)$. However, as we discussed in Sec 1, $\mathcal{G}_x$ often contains redundancy or noise. For example, given a target concept $A$, there is a semantically similar concept (e.g. a synonym) $A'$ and a noisy concept $B$ in the graph $\mathcal{G}_x$). Obviously, $A'$ will negatively impact the diversity of generations because the model may select both

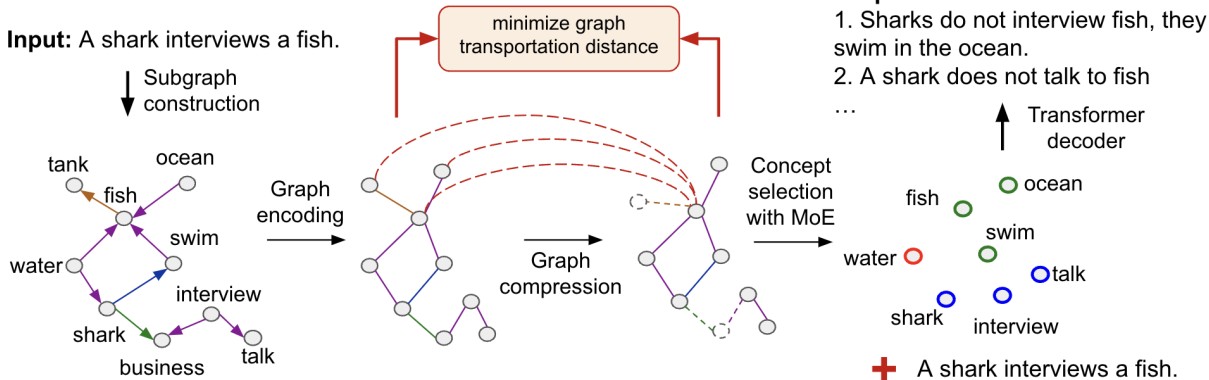

Figure 2: Overview of our approach. We retrieve a subgraph from ConceptNet for the given input sentence, compress it, and use MoE to generate diverse sentences for containing concepts from the compressed graph.

$A$ and $A'$ for generation and the semantics of the generations are similar; concept $B$ will hurt the generation quality since it is irrelevant to the context. So, a natural idea to solve the problem is to eliminate these concepts by compressing the graph.

Our method extends MoKGE (Yu et al., 2022) by compressing the retrieved external knowledge graph. The framework is illustrated in Figure 2 and described in detail subsequently. In a nutshell, it aims to identify the concepts within the KG that provide the most relevant knowledge for a particular instance. We first extract a subgraph from the KG based on the given input sentence, and encode it into a vector representation (Sec 3.1). Then, we learn a compressed graph that maintains only the most relevant concepts for the given instance (Sec 3.2). We train the model with the corresponding losses (Sec 3.3) and finally apply MoE to generate diverse outputs (Sec 3.4).

### 3.1 KG Subgraph Extraction and Encoding

The subgraph extraction and encoding follows MoKGE (Yu et al., 2022).

**Subgraph Extraction.** We first associate each input sentence with the set of concepts from the KG that match its tokens. For example, given the sentence $q =$"A shark interviews a fish" (the "query"), we extract the concepts $C_q = \{\mathtt{fish}, \mathtt{shark}, \mathtt{interview}\}$ from ConceptNet.[2] Second, we fix a radius $h$ and extract a subgraph $\mathcal{G}_q$ with node set $V_q \supseteq C_q$ from the KG such that it contains all KG nodes and edges that are up to $h = 2$ hops around the concepts in $C_q$ (e.g. shark

$\to \mathtt{swim} \to \mathtt{fish}$).

**Graph Encoding.** To obtain embeddings for the concept nodes, we apply an off-the-shelf graph encoder over the extracted subgraph (Wu et al., 2021). In our implementation, we follow Yu et al. (2022) and use the relational graph convolutional network (R-GCN; Schlichtkrull et al., 2018). R-GCN computes node representations by iteratively aggregating neighboring node representations and thereby taking the relation types into account. In this way, the final embeddings capture the structural patterns of the subgraph.

### 3.2 Differentiable Graph Compression

As we discussed before, the extracted subgraphs often contain redundancy and noise, and we aim to compress the graph and remove the irrelevant information. This introduces two challenges: (1) how to make the graph compression differentiable so that it can be trained in the context of downstream tasks; and (2) how to maintain the most important and relevant information in the compressed graph.

**Self-Attention for Concept Scoring.** Since we want to select concepts for the generation step (Sec 3.4), we can't apply differentiable pooling methods (Ying et al., 2018; Ma and Chen, 2020) and instead choose to construct a semantically meaningful subgraph containing the relevant nodes and edges. To do so, we apply self-attention and hence essentially use the features computed in the previous step as main criterion to determine the concepts' importance. Specifically, we compute self-attention scores $Z \in \mathbb{R}^{C \times 1}$ as proposed by Lee et al. (2019) using graph convolution (Kipf

---

[2]In what follows, our notation refers to KG concepts and their corresponding KG nodes interchangeably.

and Welling, 2017):

$$Z = \sigma(\tilde{D}^{-\frac{1}{2}}\tilde{A}\tilde{D}^{-\frac{1}{2}}X\Theta_{att})$$

where $\sigma$ is the non-linear activation function $tanh$; $C := |V_q|$ is the number of concept nodes in the subgraph; $\tilde{A} \in \mathbb{R}^{C\times C}$ is the adjacency matrix extended by self-connections; $\tilde{D}$ is the degree matrix of $\tilde{A}$, which is used for normalization; $X \in \mathbb{R}^{C\times F}$ is the matrix of concept embeddings obtained in the previous step, with embedding dimension $F$; and $\Theta_{att} \in \mathbb{R}^{F\times 1}$ is the parameter matrix for the self-attention scores. Given the concept scores $Z$, we consider a pre-set assignment ratio $s \in (0, 1]$, and form the *compressed graph*, $\mathcal{G}'$, by selecting $s\%$ of concept nodes. We denote $S$ as the number of concept nodes selected. In the example in Figure 2, the compressed (third) graph contains 80% of the nodes in the original subgraph.

**Optimal Transport for Regularization.** The self-attention based concept selection make the graph compressed in an differentiable way, however the attention parameters can only be trained from downstream generation tasks which cannot gurantee the compression quality as well as generalizability. Consider the case with concept $A$ and its synonym $A'$ in the retrieved graph $\mathcal{G}_q$, if $A$ is selected by the attention scores, it is highly possible $A'$ also has a high score to be selected, so the redundancy cannot be removed.

For this reason, we additionally apply optimal transport (OT; Peyré and Cuturi, 2019), a method commonly used for measuring the distance between two probability measures. Here, we regard a graph as a discrete distribution, similarly to Ma and Chen (2020), and minimize the OT distance between the original graph and its compressed version. To this end, we define an optimal transport loss between graphs. Given a $m$-node graph and a $n$-node graph, we assume they have discrete distributions $\mu = \sum_{i=1}^{m} a_i \sigma_{x_i}$ and $\nu = \sum_{j=1}^{n} b_j \sigma_{x_j}$, where $x_i$ and $x_j$ indicate the nodes, $\sigma$ is a delta function, $a = (a_1, ..., a_m)$ and $b = (b_1, ..., b_n)$ are weights of nodes (generally uniform). If we define a cost matrix $M$ whose element $M_{ij}$ indicates the transport cost from node $x_i$ to node $x_j$, then the optimal transport distance is:

$$W(\mu, \nu) = \min_{T} <T, M> \qquad (1)$$

$T \in \mathbf{R}^{m*n}$ is called a transportation plan, whose element $T_{ij}$ denotes the transportation probability

from $x_i$ to $x_j$, and it meets the requirements that $T1_n = a$, and $T^T 1_m = b$.

Once the optimal transport distance is minimized, the compressed graph is expected to keep as much information of the original graph. Thus redundant concepts will be largely removed, since involving them in the compressed graph will lead to less information kept. Take a simple example, given an original graph with nodes $\{A, A', C\}$, the subgraph with node $\{A, C\}$ should be more informative than the one with $\{A, A'\}$, and its optimal transport distance between the original graph should be smaller.

Since solving an OT problem is computationally expensive, we add an entropy regularization term $E(T) = \sum_{ij} T_{ij}(\log T_{ij} - 1)$, to allow for solving it approximately using Sinkhorn's algorithm (Cuturi, 2013) in practice, following prior work. With a hyperparameter $\gamma > 0$, the entropy-regularized loss becomes:

$$W_\gamma(\mu, \nu) = \min_{T} <T, M> -\gamma E(T) \qquad (2)$$

### 3.3 Loss Functions for Training

Following Yu et al. (2022), we train BART-base (Lewis et al., 2020) in a seq2seq architecture on the commonsense explanation generation task, with a **generation loss**, and apply a **KG concept loss** in addition. We also include an **optimal transport loss**.

**Generation Loss.** For sentence generation, we maximize the conditional probability of the target sequence $y$ given the input sequence $x$ concatenated with the selected KG concepts $c_1, c_2, ...c_S$. We utilize the standard auto-regressive cross-entropy loss as follows:

$$\mathcal{L}_g = -\sum_{t=1}^{|y|} \log P(y_t|x, c_1, c_2, ..., c_S, y_{<t})$$

where $t$ is the timestep of the actual output. In the generation step, the model auto-regressively generates the output $y$ with input $x$ and $S$ selected concepts.

**KG Concept Loss.** The effectiveness of the concept selection can be measured in terms of which of the chosen concepts appear in the output sentence $a$ (the reference answer). More specifically, we consider a regular binary cross entropy loss with targets $y_c = I(c \in V_q \cap C_a)$ for each $c \in V_q$. Here, $I(\cdot)$ represents the indicator function. and $C_a$ is

the set of concepts that are present in the output. To obtain a probability for each of the $S$ concepts in the compressed graph, we apply an MLP. The resulting loss is as follows:

$$\mathcal{L}_c = - \left( \sum_{c \in V_q \cap C_a} y_c \log P(c) + \sum_{c \in V_q - C_a} (1 - y_c) \log 1 - P(c) \right)$$

**Optimal Transport Loss.** To make the optimal transport distance differentiable, we solve Eq. 2 using the Sinkhorn's algorithm (Cuturi, 2013):

Starting with any positive vector $v^0$, we iteratively update $u$ and $v$ as follows:

$$u^{i+1} = a \oslash K v^i; v^{i+1} = b \oslash K^T u^{i+1} \quad (3)$$

where $\oslash$ is the element-wise division and $K$ is an intermediate variable derived from the cost matrix $M$: $K = \exp(-M/\gamma)$.

After $k$ steps, we arrive at the k-step result $P^k = \text{diag}(u^k) K \text{diag}(v^k)$ as an approximated optimal transportation plan, hence the optimal transport loss is approximated by

$$\mathcal{L}_t = W_\gamma^k(G, G_c) = <P^k, M> -\gamma E(P^k)$$

Altogether, our model is trained with three loss functions:

$$\mathcal{L} = \mathcal{L}_g + \alpha \mathcal{L}_c + \beta \mathcal{L}_t \quad (4)$$

where $\alpha$ and $\beta$ are hyperparameters that control the relative importance of the individual loss functions. In our experimental setup, we set both $\alpha$ and $\beta$ to a value of 0.3.

### 3.4 Diverse Generation based on MoE

To encourage more diverse outputs, we follow previous work (Shen et al., 2019; Cho et al., 2019; Yu et al., 2022) and use mixture of experts (MoE).

We use $K$ experts, where each expert is responsible for generating a unique set of KG concepts. The model is trained using hard-EM algorithm (Dempster et al., 1977). Since it is similar to (Yu et al., 2022)), we put the details in Appendix E. In Figure 2, the nodes in the 4th graph highlighted in green, red, and blue colors indicate the $K = 3$ respective experts assigned to handle different concepts. The utilization of our compressed graph version helps the model better prioritize the crucial concepts during output generation, as we demonstrate in our experiments.

## 4 Experimental Setup

### 4.1 Datasets

**ComVE** (Wang et al., 2020) was part of the SemEval 2020 commonsense validation task. Given a nonsensical sentence, the task is to generate explanations for why it doesn't make sense. The dataset contains 10k training examples and roughly 1000 examples each for test and validation. Each example comes with 3 reference output sentences. The other dataset, **α-NLG** (Bhagavatula et al., 2020), addresses the abductive commonsense reasoning task. Given a past observation and a future observation, the goal is to generate plausible explanations for what might have happened in-between. The dataset consists of 50k training examples, 1,779 validation and 3,560 test examples. Each example in the dataset includes up to 5 reference outputs.

### 4.2 Baselines

**MoE-based Methods.** **MoE-embed** (Cho et al., 2019) and **MoE-prompt** (Shen et al., 2019) produce diverse sentences by sampling different mixture components. While **MoE-embed** employs independent latent variables when generating diverse outputs, **MoE-prompt** shares the latent variable between the experts. **MoKGE** (Yu et al., 2022) is the approach that we extend by adding graph compression. It generates outputs by incorporating KG concepts on top of MoE-based methods.

**Other Methods to Improve Diversity.** To show that our method yields a sophisticated concept selection beyond regular filtering, we compare it to a simple **synonym filtering** on top of MoKGE, applied during the inference step, that yields a set of unique KG concepts for generating outputs. This baseline prevents the model from selecting similar concepts when generating the outputs. Second, we consider the common **pruning** approach, which removes irrelevant paths from the potentially noisy subgraph, following KagNet (Lin et al., 2019). To measure the quality of the path, the path is decomposed into a set of triples. Each triple is scored based on the scoring function of the knowledge graph embedding technique, TransE (Bordes et al., 2013) and the score for each path is the product of its triple scores. The threshold for pruning is a hyperparameter and set to 0.15 following Lin et al. (2019).

**Large Language Model (LLM).** Lastly, we compare to **Vicuna-13b** (Chiang et al., 2023). This

| **ComVE** | self-bleu-3 ($\Downarrow$) | self-bleu-4 ($\Downarrow$) | distinct-2 ($\Uparrow$) | entropy-4 ($\Uparrow$) | bleu-4 ($\Uparrow$) | rouge-1 ($\Uparrow$) |
|---|---|---|---|---|---|---|
| MoE, embed | $33.64_{0.2}$ | $28.21_{0.1}$ | $46.57_{0.2}$ | $9.61_{0.1}$ | $18.66_{0.5}$ | $\mathbf{43.72}_{0.2}$ |
| MoKGE, embed | $35.36_{1.1}$ | $29.71_{1.2}$ | $47.51_{0.4}$ | $9.63_{0.1}$ | $\mathbf{19.13}_{0.1}$ | $43.7_{0.1}$ |
| + SAG + OT (ours) | $\mathbf{32.19}_{0.6}$ | $\mathbf{26.28}_{0.6}$ | $\mathbf{49.05}_{0.1}$ | $\mathbf{9.69}_{0.0}$ | $19.08_{0.2}$ | $43.65_{0.3}$ |
| MoE, prompt | $33.42_{0.3}$ | $28.4_{0.3}$ | $46.93_{0.2}$ | $9.6_{0.2}$ | $18.91_{0.4}$ | $43.71_{0.5}$ |
| MoKGE, prompt | $30.93_{0.9}$ | $25.3_{1.1}$ | $48.44_{0.2}$ | $9.67_{0.2}$ | $19.01_{0.1}$ | $43.83_{0.3}$ |
| + filtering | $34.01_{0.5}$ | $28.92_{0.5}$ | $47.49_{0.9}$ | $9.64_{0.1}$ | $19.02_{0.4}$ | $43.48_{0.6}$ |
| + pruning | $33.43_{2.0}$ | $28.27_{2.2}$ | $48.26_{0.7}$ | $9.64_{0.0}$ | $18.67_{0.2}$ | $43.10_{0.3}$ |
| + SAG (ours) | $28.46_{0.8}$ | $22.81_{1.2}$ | $48.33_{0.6}$ | $9.66_{0.0}$ | $19.00_{0.6}$ | $43.80_{0.5}$ |
| + SAG + OT (ours) | $\mathbf{27.32}_{0.3}$ | $\mathbf{21.94}_{0.4}$ | $\mathbf{48.94}_{0.1}$ | $\mathbf{9.69}_{0.0}$ | $\mathbf{19.31}_{0.3}$ | $\mathbf{44.16}_{0.1}$ |
| **$\alpha$-NLG** | self-bleu-3 ($\Downarrow$) | self-bleu-4 ($\Downarrow$) | distinct-2 ($\Uparrow$) | entropy-4 ($\Uparrow$) | bleu-4 ($\Uparrow$) | rouge-1 ($\Uparrow$) |
| MoE, embed | $29.02_{1.0}$ | $24.19_{1.0}$ | $36.22_{0.3}$ | $10.84_{0.0}$ | $\mathbf{14.31}_{0.2}$ | $\mathbf{38.91}_{0.2}$ |
| MoKGE, embed | $29.17_{1.5}$ | $24.04_{1.6}$ | $38.15_{0.3}$ | $10.9_{0.1}$ | $13.74_{0.2}$ | $38.06_{0.2}$ |
| + SAG + OT (ours) | $\mathbf{24.98}_{0.2}$ | $\mathbf{19.83}_{0.2}$ | $\mathbf{38.93}_{0.3}$ | $\mathbf{10.93}_{0.0}$ | $13.06_{0.3}$ | $37.77_{0.3}$ |
| MoE, prompt | $28.05_{2.0}$ | $23.18_{1.9}$ | $36.71_{0.1}$ | $10.85_{0.0}$ | $\mathbf{14.26}_{0.3}$ | $38.78_{0.4}$ |
| MoKGE, prompt | $27.40_{2.0}$ | $22.43_{2.4}$ | $38.01_{0.6}$ | $10.88_{0.2}$ | $14.17_{0.2}$ | $38.82_{0.7}$ |
| + filtering | $31.38_{2.9}$ | $26.36_{2.8}$ | $37.95_{0.6}$ | $10.78_{0.6}$ | $13.89_{0.2}$ | $38.07_{0.1}$ |
| + pruning | $31.84_{2.3}$ | $26.72_{2.4}$ | $38.21_{0.2}$ | $10.78_{0.0}$ | $13.73_{0.1}$ | $38.01_{0.2}$ |
| + SAG (ours) | $28.49_{0.8}$ | $23.59_{0.5}$ | $38.05_{0.4}$ | $10.86_{0.0}$ | $13.41_{0.5}$ | $38.00_{0.3}$ |
| + SAG+OT (ours) | $\mathbf{23.99}_{0.7}$ | $\mathbf{18.80}_{0.6}$ | $\mathbf{39.02}_{0.7}$ | $\mathbf{10.88}_{0.0}$ | $14.21_{0.5}$ | $\mathbf{38.93}_{0.2}$ |

Table 1: Diversity and quality evaluation on ComVE and $\alpha$-NLG datasets. All experiments are run three times with different random seeds, and the standard deviations are reported in subscript.

large LM was built upon LLaMA-13b (Touvron et al., 2023), a transformer-based LM trained on trillions of tokens exclusively sourced from publicly available data. Vicuna-13b performs on par with ChatGPT (Chiang et al., 2023). We employ Vicuna-13b in a 2-shot setup (see Appendix A for the prompts).

## 4.3 Metrics

Following the same evaluation setting in previous works, we assess the performance of the generated sentences in terms of both diversity and quality.

**Pairwise Diversity.** Self-BLEU (Zhu et al., 2018) is used to evaluate how each sentence is similar to the other generated sentences based on n-gram overlap. Self-BLEU-3/4 are diversity scores based on 3/4-gram overlap. The metrics compute the average of sentence-level self-BLEU scores between all pairwise combinations of generated outputs. Hence, lower self-BLEU scores indicate a greater variety between the sentences in the set generated for each input sample.

**Corpus Diversity.** Distinct-$k$ (Li et al., 2016) is calculated by counting the number of unique $k$-grams in the generated sentence and dividing it by the total number of generated tokens, to prevent preference towards longer sentences. Additionally, we report entropy-$k$ (Zhang et al., 2018), which evaluates the evenness of the empirical n-gram dis-

tribution within the generated sentence.

**Quality.** We use standard metrics for automatic evaluation of generative tasks: BLEU (Papineni et al., 2002) and ROUGE (Lin, 2004), which are based on n-gram overlap between the generated sentences and human-written reference outputs. They assess the highest accuracy by comparing the best generated sentences to the target sentences.

## 5 Results and Discussion

**Comparison to Baselines, Table 1.** We observe similar trends for the two datasets and across the two model series, based on embedding and prompts. Overall, the differences are strongest for self-BLEU and Distinct-2, two aspects that are particularly important for diverse generation. This suggests that our model is able to reason about different possible contexts. On both datasets, our method, MoKGE+SAG+OT, outperforms the mixtures of experts by large margins in terms of diversity and, at the same time, achieves comparable or better performance in terms of quality. Note that, on ComVE, the quality differences between the best and our, second-best model are within standard deviation.

The effectiveness of our approach is especially evident from the comparison to the filtering and pruning baselines. Recall that these approaches similarly aim at better exploiting the KG by im-

proving diversity and removing noise, respectively. However, we observe a considerable decrease in diversity and nearly always also slightly in quality. This shows that *simple solutions, unrelated to the task at hand, are seemingly not able to identify the most relevant knowledge.* More specifically, for the filtering baseline, we observed that the model is unable to learn what concepts to choose for unseen data. As a result, its ability to generalize to unseen data is limited, resulting in lower diversity scores on the test data. Altogether, this demonstrates that our approach, based on the compressed graph, is effective in suppressing redundant information present in the KG and promoting other knowledge that is more relevant in the given context.

We additionally confirm that our optimal transport loss helps the model to keep the KG subgraph more coherently; see especially the $\alpha$-NLG results.

**Generation Examples, Figure 4.** Observe that MoKGE tends to generate sentences with simpler structure and fewer concepts, whereas our model employs a broader range of KG concepts. This makes the generations effectively more similar to the human-written ones, where each of the three sentences addresses a different context. We show more examples in Appendix B.

**Testing Robustness with Potentially more Redundancy and Noise, Table 2.** We created a more challenging scenario by extending the extracted subgraphs with additional, related knowledge potentially including more of both relevant and redundant information. This was done by applying COMET (Bosselut et al., 2019), a transformer that was trained to generate KG triples (i.e., entity-relation-entity tuples) based on given entity-relation pairs. The original MoKGE model seems to struggle in this scenario: its performance decreases without exception in terms of all metrics. In contrast, our approach, applied on top of MoKGE, is successful in both retaining the performance of MoKGE alone and even the improvements of MoKGE+SAG+OT.

**Comparison with Large Language Model, Table 3 & Figure 4.** We compare our method to Vicuna-13b. Most interestingly, our proposal outperforms the LLM on Distinct-2 and Entropy-4. Note that even MoKGE alone is slightly better than the LLM in these aspects, yet our method is effective in extending the gap by better exploiting the external knowledge. Figure 4 gives example

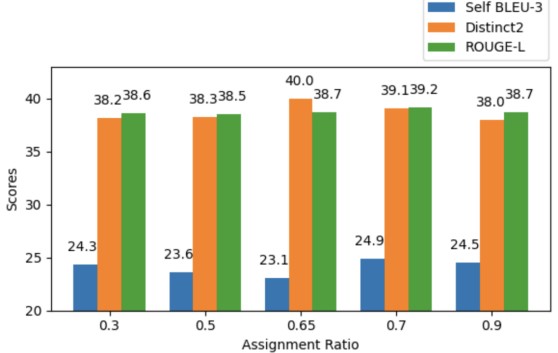

Figure 3: Self BLEU-3, Distinct-2, and ROUGE-l per assignment ratio on $\alpha$-NLG dataset. MoKGE-prompt with Self Attention and Optimal Transport is used for the experiment.

outputs and shows that the LLM is still prone to generating sentences with similar structure (e.g. "I wore a wig to ..."), as it can be seen with $\alpha$-NLG. Furthermore, while the generated sentence "I wore a wig to a party and felt great" explains the first observation "I always wondered why I loved wearing wigs", it fails to explain the second observation "I got beat up and reminded of why I shouldn't". In the ComVE task, the generated sentences are diverse in terms of both sentence structure and word usage, but the model sometimes generates sentences that are less logical, such as "Writing in a paper with an eraser is not permanent". In contrast, our approach enables MoKGE to generate a wider range of sentences that incorporate relevant concepts and enhance the interpretability of the generation process.

# 6 Analysis

**Compression Ratios, Figure 3.** This hyperparameter determines the amount of concept nodes to be kept in the compressed subgraph. Maintaining 65% of the nodes in the subgraph yields the optimal performance in terms of both diversity and quality on both datasets (see Appendix C for ComVE dataset). Interestingly, we do not observe a great negative impact on performance, even up to a ratio of 30%. This shows that ConceptNet apparently contains much information that is not necessarily beneficial for diverse generations in the context of a particular task and hence justifies our proposal.

**Unique Concepts in the Output, Appendix D.** The comparison of MoKGE and MoKGE+SAG+OT shows that MoKGE tends to generate more sentences containing 0

| ComVE | self-bleu-3 ($\Downarrow$) | self-bleu-4 ($\Downarrow$) | distinct-2 ($\Uparrow$) | entropy-4 ($\Uparrow$) | bleu-4 ($\Uparrow$) | rouge-l ($\Uparrow$) |
|---|---|---|---|---|---|---|
| MoKGE | $30.93_{0.9}$ | $25.3_{1.1}$ | $48.44_{0.2}$ | $9.67_{0.2}$ | $19.01_{0.1}$ | $43.83_{0.3}$ |
| +COMET | $32.73_{1.5}$ | $27.45_{1.8}$ | $48.32_{0.2}$ | $9.64_{0.0}$ | $18.68_{0.3}$ | $43.51_{0.4}$ |
| +COMET+SAG+OT | $\mathbf{27.23}_{1.2}$ | $\mathbf{21.68}_{1.5}$ | $\mathbf{48.65}_{0.6}$ | $\mathbf{9.68}_{0.0}$ | $\mathbf{19.38}_{0.4}$ | $\mathbf{43.99}_{0.4}$ |
| $\alpha$-**NLG** | self-bleu-3 ($\Downarrow$) | self-bleu-4 ($\Downarrow$) | distinct-2 ($\Uparrow$) | entropy-4 ($\Uparrow$) | bleu-4 ($\Uparrow$) | rouge-l ($\Uparrow$) |
| MoKGE | $27.40_{2.0}$ | $22.43_{2.4}$ | $38.01_{0.6}$ | $\mathbf{10.88}_{0.2}$ | $\mathbf{14.17}_{0.2}$ | $\mathbf{38.82}_{0.7}$ |
| +COMET | $31.41_{2.4}$ | $26.32_{2.4}$ | $37.99_{0.2}$ | $10.77_{0.1}$ | $13.87_{0.3}$ | $37.96_{0.1}$ |
| +COMET+SAG+OT | $\mathbf{25.48}_{1.0}$ | $\mathbf{21.14}_{1.3}$ | $\mathbf{38.36}_{0.3}$ | $10.84_{0.0}$ | $14.07_{0.4}$ | $38.65_{0.4}$ |

Table 2: Performance with COMET and COMET with Self Attention and Optimal Transport. MoKGE-prompt is used for experiments.

| ComVE | self-bleu-3 ($\Downarrow$) | self-bleu-4 ($\Downarrow$) | distinct-2 ($\Uparrow$) | entropy-4 ($\Uparrow$) | bleu-4 ($\Uparrow$) | rouge-l ($\Uparrow$) |
|---|---|---|---|---|---|---|
| Vicuna-13b | $\mathbf{18.10}_{0.0}$ | $\mathbf{12.74}_{0.0}$ | $48.40_{0.0}$ | $9.65_{0.0}$ | $17.65_{0.0}$ | $43.97_{0.0}$ |
| MoKGE+SAG+OT | $27.32_{0.3}$ | $21.94_{0.4}$ | $\mathbf{48.94}_{0.1}$ | $\mathbf{9.69}_{0.0}$ | $\mathbf{19.31}_{0.3}$ | $\mathbf{44.16}_{0.1}$ |
| $\alpha$-**NLG** | self-bleu-3 ($\Downarrow$) | self-bleu-4 ($\Downarrow$) | distinct-2 ($\Uparrow$) | entropy-4 ($\Uparrow$) | bleu-4 ($\Uparrow$) | rouge-l ($\Uparrow$) |
| Vicuna-13b | $33.23_{0.0}$ | $27.39_{0.0}$ | $37.97_{0.0}$ | $10.38_{0.0}$ | $\mathbf{17.30}_{0.0}$ | $\mathbf{40.58}_{0.0}$ |
| MoKGE+SAG+OT | $\mathbf{23.99}_{0.7}$ | $\mathbf{18.80}_{0.6}$ | $\mathbf{39.02}_{0.7}$ | $\mathbf{10.88}_{0.0}$ | $14.21_{0.5}$ | $38.93_{0.2}$ |

Table 3: Comparison between Vicuna-13b with 2-shot setup and MoKGE with SAG Pooling. MoKGE-prompt is used for experiments. Vicuna-13b was ran 1 time.

I always wondered why I loved wearing wigs.
[likely explanation between two events]
I got beat up and reminded of why I shouldn't.

**Human:**
1. but my parents told me it was wrong because im a male.
2. I wore a wig to school.
3. My friends caught me wearing moms wig.
4. People didn't liked when i looked like a girl.

**MoKGE:**
concepts: ['fight', 'wear', 'wig']
1. I decided to **wear** a **wig** in a **fight**.
2. One day, I decided to wear a wig.
3. I went out and got a wig for Halloween.

**Vicuna-13b:**
1. I wore a wig to a party and felt great.
2. I wore a wig to work and felt confident.
3. I wore a wig to bed and felt like a queen.

**MoKGE+SAG+OT:**
concepts: ['fight', 'wig', 'party', 'group', 'wear']
1. I wore a **wig** in front of a **group** of people.
2. I got in a **fight** one day **wearing** a wig.
3. I wore a wig to a **party** and lost all my wigs.

α-NLG

He used an eraser to write in a paper.
⇒ [explanation for sense-making]

**Human:**
1. An eraser is used to erase something written in a paper.
2. Erasers rub out write marks.
3. Eraser doesn't write.

**MoKGE:**
KG-concepts: ['paper', 'pencil', 'use', 'mark']
1. eraser is **used** to erase **pencil marks** on paper.
2. The eraser is used to erase pencil marks.
3. Eraser is used to erase pencil marks on **paper**.

**Vicuna-13b:**
1. An eraser is used to erase mistakes, not to write.
2. Writing in a paper with an eraser is not permanent.
3. Eraser is not a writing tool, it is an erasing tool.

**MoKGE+SAG+OT:**
KG-concepts: ['paper', 'pencil', 'use', 'cause', 'thing', 'mark', 'write']
1. eraser is **used** to erase **writing** on **paper**.
2. Eraser is used to erase **pencil marks** on paper.
3. You can't **write** with eraser **because** eraser is used to erase **things**.

ComVE

Figure 4: Examples of model generated sentences using MoKGE, Vicuna-13b, and MoKGE with Self Attention + Optimal Transport.

or 1 concepts only. This indicates that the lower diversity scores of MoKGE may be due to the selection of irrelevant concepts during output generation, showing the model's inability to effectively utilize them. The table shows that our method increases the numbers of KG knowledge actually used by the model and thus its success in injecting external bias into LMs.

**Human Evaluation, Table 4.** We conducted human evaluation on the outputs produced by our model MoKGE+SAG+OT and the baseline MoKGE for the $\alpha$-NLG task. We randomly se-

| Model | diversity | quality |
|---|---|---|
| MoKGE | 1.88 | 1.93 |
| MoKGE+SAG+OT | 2.10 | 2.08 |

Table 4: Human evaluation performance on 30 randomly selected $\alpha$-NLG samples.

lected 30 generations from each model. The annotation was performed by 3 researchers in the lab. We instructed the annotators to score the diversity and correctness (quality) of each generation on a scale of 0 to 3. Table 4 shows a consistent performance improvement across both diversity and quality when comparing our model to the baseline.

## 7 Conclusion

We present a differentiable graph compression algorithm that enables the model to focus on crucial information. Through experiments on two commonsense explanation generation tasks, we show that our approach not only improves the diversity but also maintains the quality of outputs. Moreover, our graph compression helps the model regain performance when new and potentially noisy information is added to graphs. Our work opens up future research in effectively selecting and incorporating symbolic knowledge into NLP models.

## Limitations

**Use of Single Word Concept.** Since ConceptNet contains mostly single words, we limit additional KG concepts to single words only. However, it can easily be extended into phrases and we leave it to future work to investigate how to effectively utilize longer phrases.

**Use of Relations.** When additional KG concepts are added to the model, we focus more on the concept nodes, not the edges. However, relation edges may provide additional insight. We leave the exploration of this for future work.

## Ethics Statement

**Data** The datasets used in our work, SemEval-2020 Task 4 Commonsense Validation and Explantation (ComVE; Wang et al., 2020) and Abductive Commonsense Reasoning ($\alpha$-NLG; Bhagavatula et al., 2020), are publicly available. The two datasets aim to produce commonsense explanations and do not include any offensive, hateful, or sexual content. The commonsense knowledge graph,

ConceptNet, was collected through crowdsourcing and may also introduce bias to our model (Mehrabi et al., 2021). However, we only use single word nodes from ConceptNet, which limits the impact of such bias.

**Models** The generative models presented in the paper are trained on a large-scale publicly available web corpus and may also bring some bias when generating sentences.

## Acknowledgements

This work was funded, in part, by the Vector Institute for AI, Canada CIFAR AI Chairs program, an NSERC discovery grant, and a research gift from AI2.

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

## A    Prompt used with Vicuna-13b

We present the prompts that we used for Vicuna-13b for ComVE (Figure 5) and $\alpha$-NLG (Figure 6).

```
# few-shot examples
< input sentence >
Give three reasons explaining why the above
sentence does not make sense:
1. < reference1 >
2. < reference2 >
3. < reference3 >
...
# target example
< input sentence >
Give three reasons explaining why the above
sentence does not make sense:
```

Figure 5: Vicuna prompt for the ComVE dataset.

```
# few-shot examples
Write three sentences that likely happened in
between the past event: < past event > and the
future event: < future event >:
1. < reference1 >
2. < reference2 >
3. < reference3 >
...
# target example
Write three sentences that likely happened in
between the past event: < past event > and the
future event: < future event >:
```

Figure 6: Vicuna prompt for the $\alpha$-NLG dataset.

| Data | Model | # of KG Concepts | | | |
|------|-------|---|---|---|----|
| | | 0 | 1 | 2 | 3<= |
| ComVE | MoKGE | 5.9 | 23.2 | 28.9 | 42.1 |
| | +SAG+OT | +0.1 | -3.1 | +1.5 | +1.0 |
| $\alpha$-NLG | MoKGE | 16.8 | 31.9 | 29.0 | 22.3 |
| | +SAG+OT | -2.0 | -1.1 | +1.7 | +1.4 |

Table 5: Comparison of models with MoKGE and MoKGE with Self Attention and Optimal Transport on the number of unique concepts in generated outputs. All KG concepts are lemmatized.

## B Additional Generation Examples

We show additional sentences generated by MoKGE and MoKGE+SAG+OT for ComVE (Figure 7) and $\alpha$-NLG (Figure 8).

## C Assignment Ratio for ComVE

We show the performance on ComVE with varying node assignment ratios in Figure 9.

## D Concept Inclusiveness

We analyze how well the model incorporates KG concepts in output generation in Table 5.

## E Mixture of Experts

Given input sentence $q$ and target sentence $y$, MoE employs a multinomial latent variable $\delta \in \{1, 2, ..., K\}$ and decomposes the marginal likelihood as:

$$P(y|x, g_x) = \sum_{\delta=1}^{K} P(\delta|x, \mathcal{G}'_x) P(y|\delta, x, \mathcal{G}'_x)$$

Each $\delta$ represents an expert, which is responsible for explaining $(x, \mathcal{G}'_x, y)$ observation.

With the above decomposition, the model minimizes the loss function (Eq.(4))

$$\nabla \log P(y|x, \mathcal{G}'_x) = \sum_{\delta=1}^{K} P(\delta|x, y, \mathcal{G}'_x) \cdot \nabla \log P(y, \delta|x, \mathcal{G}'_x)$$

and is trained using hard-EM algorithm (Dempster et al., 1977) as follows:

- E-step: choose expert $\delta^{\text{best}}$ with minimal loss.

$$\delta^{\text{best}} = \underset{\delta}{\operatorname{argmin}} - \log P(y, \delta|x, \mathcal{G}'_x)$$

- M-step: update the parameters of the chosen expert $\delta^{\text{best}}$ from E-step.

## F Hyper-parameters

We used BART-base (Lewis et al., 2020), which is built on the Transformer architecture with a 6 layer encoder-decoder. For model training, we used Adam optimizer with a batch size of 60 and a learning rate of 3e-5. For the ComVE dataset, the warmup steps are set to 5000. For the $\alpha$-NLG dataset, the weight decay is set to 0.01. For optimal transport, $\gamma$ is set to 1.0. As to the weights in the discrete distributions, $\{a_i\}$ are set evenly as $1/m$, and $\{b_j\}$ are all set as $1/n$, where $m$ and $n$ are number of nodes in the graphs.

A bra is for shaving a mans face.
⇒ [explanation for sense-making]

I like to buy clothes in the gym.
⇒ [explanation for sense-making]

**Human:**
1. A bra is meant for a woman's chest, and can not shave hair.
2. A bra is worn on a womens chest.
3. Bra doesn't has a blades.

**MoKGE:**
concepts: ['face', 'man', 'shaving', 'bra']
1. A **bra** is not for **shaving** a **mans** face.
2. **bra** is not for **shaving** a **man's** face.
3. **Bra** is not meant for **shaving** a **mans** face.

**MoKGE+SAG+OT:**
concepts: ['dress', 'face', 'use', 'man', 'bra', 'shave']
1. **Bra** is not for **shaving** a **mans face**, it is for **dressing** up.
2. A **bra** is not for **shaving** a **mans** face.
3. **Men** do not **shave** their **faces using bras**. **bras** are not used for **shaving**.

**Human:**
1. The gym does not sell clothes.
2. Gyms don't usually sell clothing.
3. Gym is a place where we cannot buy clothes.

**MoKGE:**
concepts: ['sell', 'buy', 'gym']
1. You can't **buy** clothes in the **gym**, it's not clothing.
2. Gym doesn't have clothes to **buy** in the **gym**.
3. gyms don't **sell** clothes, they **sell** shoes.

**MoKGE+SAG+OT:**
concepts: ['store', 'sell', 'buy', 'workout', 'gym']
1. A gym doesn't **sell** clothes. You go to the **gym** to **workout**.
2. You can't **buy** clothes in the **gym**. You go to the **store**.
3. **Gyms** don't **sell** clothes, they **sell** shoes.

ComVE

Figure 7: Additional examples of model generated sentences with MoKGE and MoKGE+SAG+OT on ComVE dataset.

Kelly decided she wanted to paint some plaster.
[likely explanation between two events]
Kelly had to redo it.

**Human:**
1. However, she realized she did it wrong.
2. Kelly applied the plaster wrong.
3. She finished her project, but then the dog ran over it and ruined it.

**MoKGE:**
concepts: ['turn', 'paint']
1. Kelly **painted** it badly and it **turned** out badly.
2. Kelly **painted** it badly. She did not realize how bad it looked.
3. She did not do it correctly and it **turned** out badly.

**MoKGE+SAG+OT:**
concepts: ['good', 'white', 'plaster', 'wall', 'paint']
1. She **painted** it badly and it didn't look **good**.
2. Kelly **painted** the **walls** with **white** paint. It looked bad.
3. Kelly did not like the look of the **plaster**.

Roy loved everything about rice.
[likely explanation between two events]
Now Roy produces millions of pounds of rice and has sales worldwide!.

**Human:**
1. Roy begin growing his own rice.
2. Roy decided to become a rice farmer.
3. Roy decided to grow rice.
4. Roy started growing his own rice and selling it.
5. Roy wanted to start producing rice.

**MoKGE:**
concepts: ['company', 'rice', 'farm']
1. Roy decided to start his own **rice company** with his passion.
2. Roy decided to start his own **rice farm** with his rice beans.
3. He decided to start his own **rice company** with his passion.

**MoKGE+SAG+OT:**
concepts: ['company', 'money', 'production', 'rice', 'farm']
1. Roy's passion got him into the field of **rice production**.
2. Roy started his own **rice farm**. Roy learned everything.
3. Roy decided to start his own **rice company** with his own **money**.

α-NLG

Figure 8: Additional examples of model generated sentences with MoKGE and MoKGE+SAG+OT on α-NLG dataset.

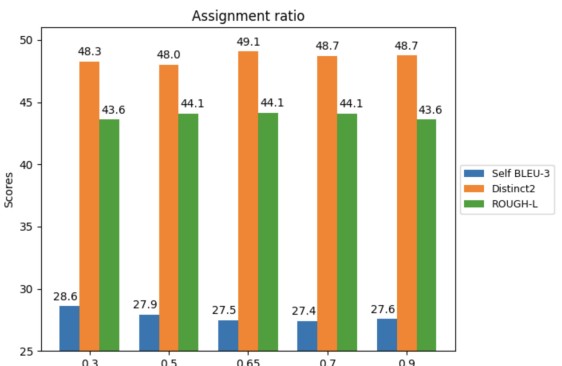

Figure 9: Self BLEU-3, Distinct-2, and ROUGE-l per
assignment ratio on ComVE dataset.