# OpenReview forum: "Knowledge Graph Compression Enhances Diverse Commonsense Generation"
_EMNLP/2023/Conference — EMNLP 2023 Main_

### Official Review · Reviewer_1hNR · 2023-07-24

**Typos Grammar Style And Presentation Improvements:** None.
**Soundness:** 4

**Excitement:**

4: Strong: This paper deepens the understanding of some phenomenon or lowers the barriers to an existing research direction.

**Missing References:**

Murphy, G. (2004). The big book of concepts. MIT press.

He, M., Fang, T., Wang, W., & Song, Y. (2022). Acquiring and modelling abstract commonsense knowledge via conceptualization. arXiv preprint arXiv:2206.01532.

Wang, W., Fang, T., Xu, B., Bo, C. Y. L., Song, Y., & Chen, L. (2023). CAT: A contextualized conceptualization and instantiation framework for commonsense reasoning. arXiv preprint arXiv:2305.04808.

Allaway, E., Hwang, J. D., Bhagavatula, C., McKeown, K., Downey, D., & Choi, Y. (2022). Penguins Don't Fly: Reasoning about Generics through Instantiations and Exceptions. arXiv preprint arXiv:2205.11658.

Wang, W., Fang, T., Ding, W., Xu, B., Liu, X., Song, Y., & Bosselut, A. (2023). CAR: Conceptualization-Augmented Reasoner for Zero-Shot Commonsense Question Answering. arXiv preprint arXiv:2305.14869.

Liu, J., Chen, T., Wang, C., Liang, J., Chen, L., Xiao, Y., ... & Jin, K. (2022). VoCSK: Verb-oriented commonsense knowledge mining with taxonomy-guided induction. Artificial Intelligence, 310, 103744.

**Paper Topic And Main Contributions:**

This paper proposes to use a differentiable graph compression algorithm to extract salient and relevant knowledge from a commonsense knowledge graph for generating diverse commonsense and abductive explanations. The compressed graph is then used to generate diverse explanations using a mixture of expert models (MoE). The authors evaluate their approach on two tasks: the ComVE and abductive NLG tasks. Empirical results show that their approach outperforms baseline methods in terms of generated quality and diversity of generated explanations.

The primary contribution of this paper is the development of the CSKG compression algorithm, which represents a compelling NLP engineering experiment.

**Questions For The Authors:**

Please find my comments above. I will consider raising my scores if the authors address the issues I have raised in their rebuttal.

**Reasons To Accept:**

The idea of compressing commonsense knowledge graphs is a novel and well-motivated approach. Although the authors developed this method with the explicit purpose of commonsense generations, I believe it has the potential to be applied to other tasks, such as commonsense QA and multi-hop reasoning.

The experiments conducted in this study are comprehensive, and the results demonstrate good performance. It is commendable that the authors not only analyzed the overall performance on two tasks but also demonstrated their improvements on COMET and included the comparison against a large language model. These findings have significant implications.

The paper is well-written and presents clear details, enabling readers to understand the methodology and results of the study. The case studies and figures provided are particularly helpful in illustrating the authors' points.

**Reasons To Reject:**

Overall, I find the method and experiments in this paper to be generally sound, and I am leaning towards accepting it. However, I do have some suggestions and questions that I would like to bring up.

Firstly, I believe that the introduction could be significantly improved. Currently, there is a large leap from discussing CSKG to concept-level CSKG, which is difficult for me to follow. I cannot help wondering why you insist on using ConceptNet but not other CSKGs such as ATOMIC or GLUCOSE. The reason seems to be, in lines 38-39, the authors argue that free-form CSKGs, such as ATOMIC, represent nodes in natural language, which adds flexibility but also introduces redundancy and noise. While this is a key point that supports the use of ConceptNet, it is not always true and requires further evidence to be convincing. Instead of relying solely on previous works, which may make the paper seem incremental, I suggest providing a more compelling motivation for the study. Specifically, it highlights the importance of concept-level understanding for realizing commonsense reasoning and pointing out the gap in the literature where previous works only focused on conceptualizing or instantiating commonsense knowledge with concepts but failed to consider their relevance or plausibility in the path of reasoning towards a specific contextual task. By doing so, the paper would have a stronger foundation and flow more smoothly. I have attached some references below for your consideration.

Another suggestion is to include a comparison to the performances of the original COMET (from ATOMIC2020) as another baseline for evaluation. This would provide additional context and help readers better understand the strengths of the proposed method.

Finally, I recommend improving the readability of Figure 3 by enlarging the font and possibly placing the legend at the top or bottom of the figure. Alternatively, placing the numbers inside the bars could also make it easier to read.

========================

After rebuttal: it seems that I have misunderstood the content regarding my second reason-to-reject. The authors responded to my other questions adequately. I raised my scores.

**Reproducibility:**

4: Could mostly reproduce the results, but there may be some variation because of sample variance or minor variations in their interpretation of the protocol or method.

**Reviewer Confidence:**

4: Quite sure. I tried to check the important points carefully. It's unlikely, though conceivable, that I missed something that should affect my ratings.

---

> ### Author Rebuttal · Authors · 2023-08-29
>
> Thank you for the very detailed feedback! We would love to address additional questions during the discussion period if anything is unclear.
>
> **Why not other CSKGs? & Improve paper flow in the introduction**
>
> Our focus is indeed on evaluating the impact of reasoning for concept selection. To see that best, we chose the most straightforward scenario, using ConceptNet, which is also the one that allows for direct comparison with the most closely related works. In this context, we apply COMET (see also our response to the comment below), to highlight our approach’s usability in a very realistic scenario, that is, by including multiple and potentially redundant data sources. We did not intend to cause confusion or recommend a particular KG. This is good feedback and we made the motivation more clear now. Thank you also for the additional references and we will add them to the revised version!
>
> **Include a comparison to the performances of the original COMET (from ATOMIC2020) as another baseline for evaluation.**
>
> We are unsure if we correctly understand this suggestion. COMET is not designed to generate multiple diverse sentences involving various concepts, so we would need to adapt it considerably for a direct comparison. Alternatively, we could replace ConceptNet with ATOMIC and use this as an external KG. This is indeed an interesting experiment to learn more about the impact of different KGs, but it requires some adaptation. Since our study’s goal is to evaluate our approach in comparison to the related works, we didn’t consider it directly. But please note that the results in Table 2 (MoKGE vs. MoKGE+COMET vs. MoKGE+COMET+SAG+OT) do include COMET and the knowledge graph expanded from COMET can be considered as another KG.
>
> **Improve the readability of Figure 3 by enlarging the font and possibly placing the legend at the top or bottom of the figure.**
>
> We will update Figure 3 in the revised version.

---

### Official Review · Reviewer_BbGf · 2023-08-04

**Soundness:** 2

**Excitement:**

3: Ambivalent: It has merits (e.g., it reports state-of-the-art results, the idea is nice), but there are key weaknesses (e.g., it describes incremental work), and it can significantly benefit from another round of revision. However, I won't object to accepting it if my co-reviewers champion it.

**Paper Topic And Main Contributions:**

The work aims to solve the task of commonsense generation via commonsense KG. Given a sentence, the work first extracts from KG a local graph containing potentially relevant concepts. To filter irrelevant concepts, the work adopts self-attention to score the concepts. To ensure quality, the work adopts optimal transport to preserve the most information of the original graph. The final selected concepts serve as additional inputs to a seq2seq model for generating the explanation. Results on two datasets demonstrate the effectiveness of the proposed method.

**Questions For The Authors:**

A. Can you explain 1) what training signal tells which concept we should choose and 2) how this signal is back-propagated to your attention network?

**Reasons To Accept:**

1.	Obtaining contextualized knowledge is an important question which may benefit not just commonsense generation but broadly language understanding and reasoning.
2.	The work explores some interesting ideas for pruning the graph while maintaining the quality.
3.	Extensive experiments and analyses are conducted to verify the effectiveness of the proposed method.

**Reasons To Reject:**

1.	The work invests the major efforts in selecting the concepts. However, it seems at the end, we still rely on the knowledge encoded in LM (e.g., a fish cannot talk) to solve the task. This raises the question of whether it is still necessary to select concepts for LM in the first place. If LM contains the required knowledge, I do not see a problem for LM to figure out the concepts by itself.
2.	In the proposed method, it is unclear how the training objectives would guide the model to select relevant concepts (and also how the training signals are back-propagated). It seems that the work assumes that whether the concept appears in the final sentence is indicative of what is a good concept. But I think many concepts that do not appear in the final sentence are also important as they introduce background knowledge (potentially multi-hop).

**Reproducibility:**

3: Could reproduce the results with some difficulty. The settings of parameters are underspecified or subjectively determined; the training/evaluation data are not widely available.

**Reviewer Confidence:**

3: Pretty sure, but there's a chance I missed something. Although I have a good feel for this area in general, I did not carefully check the paper's details, e.g., the math, experimental design, or novelty.

---

> ### Author Rebuttal · Authors · 2023-08-29
>
> Thank you for the thoughtful comments! We would love to address additional questions during the discussion period if anything is unclear.
>
> **Is it still necessary to select concepts for LM in the first place. If LM contains the required knowledge, I do not see a problem for LM to figure out the concepts by itself. If LM contains the required knowledge, I do not see a problem for LM to figure out the concepts by itself.**
>
> We intended to address this question in our LLM comparison (Table 3 and Figure 4), given that it is indeed likely that similar KG knowledge is included in the pretraining data. Our method’s focus is particularly on diversifying the output generations. Interestingly, our evaluation shows clearly that the LLM, Vicuna-13B in a 2-shot setup, cannot beat our approach in this respect. It fails to fully use its knowledge and generate diverse sentences or better-quality sentences. More generally, hallucination and lack of commonsense are still well-known problems of LLMs nowadays, so they need to be made more focused. Our research targets this direction by focusing on the aspect of learning to select KG knowledge to diversify generation.
>
> **The work assumes that whether the concept appears in the final sentence is indicative of what is a good concept. What training signal tells which concept we should choose and how this signal is back-propagated to your attention network.**
>
> That’s a good question and we will clarify this in the paper. We don’t assume that only concepts that are included in the generated sentence are helpful. Instead, the final sentence provides indirect supervision to select the most relevant concepts (i.e., the sentence may contain concepts X and Y which are connected in the KG through concept Z which is not included in the sentence). We choose the relevant nodes based on self-attention scores and then apply optimal transport, also minimizing duplicate concepts, both of which are trained.

---

### Official Review · Reviewer_wZ5V · 2023-08-04

**Soundness:** 4

**Excitement:**

3: Ambivalent: It has merits (e.g., it reports state-of-the-art results, the idea is nice), but there are key weaknesses (e.g., it describes incremental work), and it can significantly benefit from another round of revision. However, I won't object to accepting it if my co-reviewers champion it.

**Paper Topic And Main Contributions:**

The main work of this paper is compressing the commonsense knowledge graph to facilitate a better generation of commonsense explanations. The framework of this article extends from MoKGE and utilizes self-attention scores to select highly relevant entities in the subgraph. It further employs transport loss to eliminate semantically similar duplicate entities. This work compares its method on two datasets with previous models, and notably, it also conducts comparisons with a large language model. Additionally, the paper presents some case studies to demonstrate the effectiveness of the proposed method.

**Questions For The Authors:**

Question A: The prompts and inputs used for the large language model do not contain concepts, which may lead to an unfair comparison. The given concepts could have a significant effect on the generated results. If suitable concepts are identified for the large model using the method in the paper and incorporated into the prompts, would it lead to better reasoning performance?

Question B: Do the authors think the quality of the knowledge graph would affect the quality of generation? For example, ConceptNet is constructed via crowdsourcing and may contain biases.

**Reasons To Accept:**

1. This work proposes a differentiable graph compression algorithm, which can automatically identify and retain key conceptual information relevant to a specific task.
2. This work utilizes self-attention and optimal transport regularization to reduce redundant and irrelevant information, which can lead to better generation of outputs.
3. Through the proposed method in the paper, better results are demonstrated across the three metrics of pairwise diversity, corpus diversity, and quality.

**Reasons To Reject:**

1. This work does not take into account the relations when training concept embeddings. Relations are an essential component that needs to be considered in the knowledge graph.
2. This work lacks human evaluation. The current evaluation relies primarily on automated metrics. Human evaluation can better judge the quality and logicality of the generated outputs. MoE and MoKGE both conducted human evaluations.
3. The authors propose that their generation approach can be applied to other NLP tasks, but do not provide concrete descriptions or demonstrations of this in the paper.

**Reproducibility:**

3: Could reproduce the results with some difficulty. The settings of parameters are underspecified or subjectively determined; the training/evaluation data are not widely available.

**Reviewer Confidence:**

4: Quite sure. I tried to check the important points carefully. It's unlikely, though conceivable, that I missed something that should affect my ratings.

---

> ### Author Rebuttal · Authors · 2023-08-29
>
> Thank you for recognizing our contributions and for pointing out critical discussion points! We have made those more clear in the paper now.
>
> **This work does not take into account the relations when training concept embeddings**
>
> This seems to be a misunderstanding. In fact, the relational graph convolutional network (R-GCN) (see Sec 3.1), which is commonly used in the KG context [1] and which we use for learning concept embeddings, takes the relations in the neighborhoods of the concept nodes into account very directly. In this way, they are also considered in both our losses, the KG concept loss and optimal transport loss. We could add the relations explicitly, but since R-GCN has been proven generally, we resorted to this more straightforward solution.
>
> [1] Modeling relational data with graph convolutional networks. Michael Schlichtkrull et al. In The Semantic Web: 15th International Conference, ESWC 2018.
>
> **Missing human evaluation**
>
> This is a very fair point. We have tried our best to consider a broad range of metrics as suggested in prior works, covering pairwise diversity, corpus diversity, and quality. Overall, the metrics capture diversity - our main objective - well. We totally agree about the value of human evaluation to include more subtle logic. We now ran a small human evaluation with 30 random samples from alpha-NLG. The samples are randomly permuted and put on M-Turk with their model information removed. The annotators are asked to score the diversity and correctness/quality from 0 to 3. Due to the short time, we only have two annotators and the results are as follows:
>
> diversity score: baseline avg: 1.81, our model avg: 2.03
>
> quality score: baseline avg: 1.87, our model avg: 2.05.
>
> This hints at a consistent performance improvement with our model in terms of both diversity and quality metrics. We will extend this initial study and add more human evaluation results in the updated version, but the conclusion is supposed to be the same based on current observations.
>
> **No concrete descriptions or demonstrations of applications of the approach in the paper.**
>
> Our approach is particularly applicable to tasks that benefit from additional, external commonsense knowledge, such as CommonsenseQA, CommonGen, and multihop reasoning over commonsense KGs. In the paper, we focus on commonsense generation, we made this more clear now. But we added a more general discussion as an outlook.
>
>
> **LLM does not contain concepts, may leading to unfair comparison. If suitable concepts are identified for llm, would it lead to better reasoning performance?**
>
> Our intention in the LLM experiment was to compare to a “vanilla” LLM in a realistic scenario, a standard prompting with instructions in a few-shot setup. It is a valid, but different research question if we also can improve LLM performance by combining an LLM with the components of our approach (such as keyword extraction, path extraction, etc.). However, we assume that we see less direct impact here since most of these models were likely trained on knowledge similar to the one in available KGs. For this reason, we also don’t consider the comparison to be unfair.
>
> **Do the authors think the quality of the knowledge graph would affect the quality of generation?**
>
> This is actually the point we intend to analyze in our experiment in Table 2, where we add potential noise generated by COMET. We made this more clear now. Generally, the quality of the KG will likely affect models using the KG out of the box, and we can see this with MoKGE, it is strongly impacted by the noise. In contrast, our adaptive, supervised approach, especially the OT part, is able to prune noise and irrelevant knowledge. Hence it is rather robust.

---

### Meta-Review · Area_Chair_f4Go · 2023-09-19

**Recommendation:** 4

**Metareview:**

This work aims to facilitate a better generation of commonsense explanations using compressed KGs. In the proposed framework, self-attention scores to select highly relevant entities in the subgraph of the KG. The experimental results on two datasets show that the proposed method can outperform conventional baselines in diversity while keeping the comparable or better relevance to the references. Reviewer wZ5V and 1hNR gave positive scores, whereas Reviewer BbGf pointed out the unnecessity of selecting concepts from the KG for LM due to its knowledge obtained in pretraining. Considering that the authors addressed the question by referring to Table 3 and Figure 4, we can judge that there is no major problem in accepting this paper.

---

### Decision · Program_Chairs · 2023-10-07

**Decision:**

Accept-Main

**Comment:**

This work aims to facilitate a better generation of commonsense explanations using compressed KGs. In the proposed framework, self-attention scores to select highly relevant entities in the subgraph of the KG. The experimental results on two datasets show that the proposed method can outperform conventional baselines in diversity while keeping the comparable or better relevance to the references. Reviewer wZ5V and 1hNR gave positive scores, whereas Reviewer BbGf pointed out the unnecessity of selecting concepts from the KG for LM due to its knowledge obtained in pretraining. Considering that the authors addressed the question by referring to Table 3 and Figure 4, we can judge that there is no major problem in accepting this paper.